# ADAPTIVE NETWORK SPARSIFICATION VIA DEPENDENT VARIATIONAL BETA-BERNOULLI DROPOUT

## ABSTRACT

While variational dropout approaches have been shown to be effective for network sparsification, they are still suboptimal in the sense that they set the dropout rate for each neuron without consideration of the input data. With such input-independent dropout, each neuron is evolved to be generic across inputs, which makes it difficult to sparsify networks without accuracy loss. To overcome this limitation, we propose adaptive variational dropout whose probabilities are drawn from sparsity-inducing beta-Bernoulli prior. It allows each neuron to be evolved either to be generic or specific for certain inputs, or dropped altogether. Such input-adaptive sparsity-inducing dropout allows the resulting network to tolerate larger degree of sparsity without losing its expressive power by removing redundancies among features. We validate our dependent variational beta-Bernoulli dropout on multiple public datasets, on which it obtains significantly more compact networks than baseline methods, with consistent accuracy improvements over the base networks.

## 1 INTRODUCTION

One of the main obstacles in applying deep learning to large-scale problems and low-power computing systems is the large number of network parameters, as it can lead to excessive memory and computational overheads. To tackle this problem, researchers have explored network sparsification methods to remove unnecessary connections in a network, which is implementable either by weight pruning (Han et al., 2016) or sparsity-inducing regularizations Wen et al. (2016).

Recently, variational Bayesian approaches have shown to be useful for network sparsification, outperforming non-Bayesian counterparts. They take a completely different approach from the conventional methods that either uses thresholding or sparsity-inducing norms on parameters, and uses well-known dropout regularization instead. Specifically, these approaches use variational dropout Kingma et al. (2015) which adds in multiplicative stochastic noise to each neuron, as a means of obtaining sparse neural networks. Removal of unnecessary neurons could be done by either setting the dropout rate individually for each neuron with unbounded dropout rate Molchanov et al. (2017) or by pruning based on the signal-to-noise ratio Neklyudov et al. (2017).

While these variational dropout approaches do yield compact networks, they are suboptimal in that the dropout rate for each neuron is learned completely independently of the given input data and labels. With input-independent dropout regularization, each neuron has no choice but to encode generic information for all possible inputs, since it does not know what input and tasks it will be given at evaluation time, as each neuron will be retained with fixed rate regardless of the input. Obtaining high degree of sparsity in such as setting will be difficult as dropping any of the neurons will result in information loss. For maximal utilization of the network capacity and thus to obtain a more compact model, however, each neuron should be either irreplaceably generic and used by all tasks, or highly specialized for a task such that there exists minimal redundancy among the learned representations. This goal can be achieved by adaptively setting the dropout probability for each input, such that some of the neurons are retained with high probability only for certain types of inputs and tasks.

To this end, we propose a novel input-dependent variational dropout regularization for data and task-dependent network sparsification. We first propose *beta-Bernoulli dropout* that learns to set dropout rate for each individual neuron, by generating the dropout mask from beta-Bernoulli prior, and show how to train it using variational inference. This dropout regularization is a proper way of obtaining a Bayesian neural network and also sparsifies the network, since beta-Bernoulli distribution

is a sparsity-inducing prior. Then, we propose dependent beta-Bernoulli dropout, which is an input-dependent version of our variational dropout regularization.

Such adaptive regularization has been utilized for general network regularization by a non-Bayesian and non-sparsity-inducing model Ba & Frey (2013); yet, the increased memory and computational overheads that come from learning additional weights for dropout mask generation made it less appealing for generic network regularization. In our case of network sparsification, however, the overheads at training time is more than rewarded by the reduced memory and computational requirements at evaluation time, thanks to the high degree of sparsification obtained in the final output model.

We validate our dependent beta-Bernoulli variational dropout regularizer on multiple public datasets for network sparsification performance and prediction error, on which it obtains more compact network with substantially reduced prediction errors, when compared with both the base network and existing network sparsification methods. Further analysis of the learned dropout probability for each unit reveals that our input-adaptive variational dropout approach generates a clearly distinguishable dropout mask for each task, thus enables each task to utilize different sets of neurons for their specialization.

Our contribution in this paper is threefold:

- We propose beta-Bernoulli dropout, a novel dropout regularizer which learns to generate Bernoulli dropout mask for each neuron with sparsity-inducing prior, that obtains high degree of sparsity without accuracy loss.

- We further propose dependent beta-Bernoulli dropout, which yields significantly more compact network than input-independent beta-Bernoulli dropout, and further perform runtime pruning for even less computational cost.

- Our beta-Bernoulli dropout regularizations provide novel ways to implement a sparse Bayesian Neural Network, and we provide a variational inference framework for learning it.

## 2 RELATED WORK

Deep neural networks are known to be prone to overfitting, due to its large number of parameters. Dropout Srivastava et al. (2014) is an effective regularization that helps prevent overfitting by reducing coadaptations of the units in the networks. During dropout training, the hidden units in the networks are randomly dropped with fixed probability $p$, which is equivalent to multiplying the Bernoulli noises $z \sim \text{Ber}(1-p)$ to the units. It was later found that multiplying Gaussian noises with the same mean and variance, $z \sim \mathcal{N}(1, \frac{p}{1-p})$, works just as well or even better Srivastava et al. (2014).

Dropout regularizations generally treat the dropout rate $p$ as a hyperparameter to be tuned, but there have been several studies that aim to automatically determine proper dropout rate. Kingma et al. (2015) propose to determine the variance of the Gaussian dropout by stochastic gradient variational Bayes. Generalized dropout Srinivas & Babu (2016) places a beta prior on the dropout rate and learn the posterior of the dropout rate through variational Bayes. They showed that by adjusting the hyperparameters of the beta prior, we can obtain several regularization algorithms with different characteristics. Our beta-Bernoulli dropout is similar to one of its special cases, but while they obtain the dropout estimates via point-estimates and compute the gradients of the binary random variables with biased heuristics, we approximate the posterior distribution of the dropout rate with variational distributions and compute asymptotically unbiased gradients for the binary random variables.

Ba et al. Ba & Frey (2013) proposed adaptive dropout (StandOut), where the dropout rates for each individual neurons are determined as function of inputs. This idea is similar in spirit to our dependent beta-Bernoulli dropout, but they use heuristics to model this function, while we use proper variational Bayesian approach to obtain the dropout rates. One drawback of their model is the increased memory and computational cost from additional parameters introduced for dropout mask generation, which is not negligible when the network is large. Our model also requires additional parameters, but with our model the increased cost at training time is rewarded at evaluation time, as it yields a significantly sparse network than the baseline model as an effect of the sparsity-inducing prior.

Recently, there has been growing interest in structure learning or sparsification of deep neural networks. Han et al. Han et al. (2016) proposed a strategy to iteratively prune weak network weights for efficient computations, and Wen et al. Wen et al. (2016) proposed a group sparsity learning

algorithm to drop neurons, filters or even residual blocks in deep neural networks. In Bayesian learning, various sparsity inducing priors have been demonstrated to efficiently prune network weights with little drop in accuracies Molchanov et al. (2017); Louizos et al. (2017); Neklyudov et al. (2017); Louizos et al. (2018). In the nonparametric Bayesian perspective, Feng et al. Feng & Darrell (2015) proposed IBP based algorithm that learns proper number of channels in convolutional neural networks using the asymptotic small-variance limit approximation of the IBP. While our dropout regularizer is motivated by IBP as with this work, our work is differentiated from it by the input-adaptive adjustments of dropout rates that allow each neuron to specialize into features specific for some subsets of tasks.

## 3 BACKGROUNDS

### 3.1 BAYESIAN NEURAL NETWORKS AND STOCHASTIC GRADIENT VARIATIONAL BAYES

Suppose that we are given a neural network $\mathbf{f}(\mathbf{x}; \mathbf{W})$ parametrized by $\mathbf{W}$, a training set $\mathcal{D} = \{(\mathbf{x}_n, \mathbf{y}_n)\}_{n=1}^N$, and a likelihood $p(\mathbf{y}|\mathbf{f}(\mathbf{x}; \mathbf{W}))$ chosen according to the problem of interest (e.g., the categorical distribution for a classification task). In Bayesian neural networks, the parameter $\mathbf{W}$ is treated as a random variable drawn from a pre-specified prior distribution $p(\mathbf{W})$, and the goal of learning is to compute the posterior distribution $p(\mathbf{W}|\mathcal{D})$:

$$p(\mathbf{W}|\mathcal{D}) \propto p(\mathbf{W}) \prod_{n=1}^N p(\mathbf{y}_n|\mathbf{f}(\mathbf{x}_n; \mathbf{W})). \tag{1}$$

When a novel input $\mathbf{x}_*$ is given, the prediction $\mathbf{y}_*$ is obtained as a distribution, by mixing $\mathbf{W}$ from $p(\mathbf{W}|\mathcal{D})$ as follows:

$$p(\mathbf{y}_*|\mathbf{x}_*, \mathcal{D}) = \int p(\mathbf{y}_*|\mathbf{f}(\mathbf{x}_*; \mathbf{W}))p(\mathbf{W}|\mathcal{D})\mathrm{d}\mathbf{W}. \tag{2}$$

Unfortunately, $p(\mathbf{W}|\mathcal{D})$ is in general computationally intractable due to computing $p(\mathcal{D})$, and thus we resort to approximate inference schemes. Specifically, we use variational Bayes (VB), where we posit a variational distribution $q(\mathbf{W}; \boldsymbol{\phi})$ of known parametric form and minimize the KL-divergence between it and the true posterior $p(\mathbf{W}|\mathcal{D})$, $\mathrm{D}_{\mathrm{KL}}[q(\mathbf{W}; \boldsymbol{\phi})\|p(\mathbf{W}|\mathcal{D})]$. It turns out that minimizing $\mathrm{D}_{\mathrm{KL}}[q(\mathbf{W}; \boldsymbol{\phi})\|p(\mathbf{W}|\mathcal{D})]$ is equivalent to maximizing the evidence lower-bound (ELBO),

$$\mathcal{L}(\boldsymbol{\phi}) = \sum_{n=1}^N \mathbb{E}_q[\log p(\mathbf{y}_n|\mathbf{f}(\mathbf{x}_n; \mathbf{W}))] - \mathrm{D}_{\mathrm{KL}}[q(\mathbf{W}; \boldsymbol{\phi})\|p(\mathbf{W})], \tag{3}$$

where the first term measures the expected log-likelihood of the dataset w.r.t. $q(\mathbf{W}; \boldsymbol{\phi})$, and the second term regularizes $q(\mathbf{W}; \boldsymbol{\phi})$ so that it does not deviate too much from the prior. The parameter $\boldsymbol{\phi}$ is learned by gradient descent, but these involves two challenges. First, the expected likelihood is intractable in many cases, and so is its gradient. To resolve this, we assume that $q(\mathbf{W}; \boldsymbol{\phi})$ is reparametrizable, so that we can obtain i.i.d. samples from $q(\mathbf{W}; \boldsymbol{\phi})$ by computing differentiable transformation of i.i.d. noise (Kingma & Welling, 2014; Rezende et al., 2014) as $\boldsymbol{\varepsilon}^{(s)} \sim r(\boldsymbol{\varepsilon}), \mathbf{W}^{(s)} = \mathcal{T}(\boldsymbol{\varepsilon}^{(s)}; \boldsymbol{\phi})$. Then we can obtain a low-variance unbiased estimator of the gradient, namely

$$\nabla_{\boldsymbol{\phi}} \mathbb{E}_q[\log p(\mathbf{y}_n|\mathbf{f}(\mathbf{x}_n; \mathbf{W}))] \approx \frac{1}{S} \sum_{s=1}^S \nabla_{\boldsymbol{\phi}} \log p(\mathbf{y}_n|\mathbf{f}(\mathbf{x}_n; \mathcal{T}(\boldsymbol{\varepsilon}^{(s)}; \boldsymbol{\phi}))). \tag{4}$$

The second challenge is that the number of training instances $N$ may be too large, which makes it impossible to compute the summation of all expected log-likelihood terms. Regarding on this challenge, we employ the stochastic gradient descent technique where we approximate with the summation over a uniformly sampled mini-batch $B$,

$$\sum_{n=1}^N \nabla_{\boldsymbol{\phi}} \mathbb{E}_q[\log p(\mathbf{y}_n|\mathbf{f}(\mathbf{x}_n; \mathbf{W}))] \approx \frac{N}{|B|} \sum_{n \in B} \nabla_{\boldsymbol{\phi}} \mathbb{E}_q[\log p(\mathbf{y}_n|\mathbf{f}(\mathbf{x}_n; \mathbf{W}))]. \tag{5}$$

Combining the reparametrization and the mini-batch sampling, we obtain an unbiased estimator of $\nabla_{\boldsymbol{\phi}}\mathcal{L}(\boldsymbol{\phi})$ to update $\boldsymbol{\phi}$. This procedure, often referred to as stochastic gradient variational Bayes (SGVB) Kingma & Welling (2014), is guaranteed to converge to local optima under proper learning-rate scheduling.

### 3.2 Latent feature models and Indian Buffet Processes

In latent feature model, data are assumed to be generated as combinations of latent features:

$$\mathbf{d}_n = f(\mathbf{W}\mathbf{z}_n) = f\left(\sum_{k=1}^{K} z_{n,k}\mathbf{w}_k\right), \tag{6}$$

where $z_{n,k} = 1$ means that $\mathbf{d}_n$ possesses the $k$-th feature $\mathbf{w}_k$, and $f$ is an arbitrary function.

The Indian Buffet Process (IBP) Griffiths & Ghahramani (2005) is a generative process of binary matrices with infinite number of columns. Given $N$ data points, IBP generates a binary matrix $\mathbf{Z} \in \{0,1\}^{N \times K}$ whose $n$-th row encodes the feature indicator $\mathbf{z}_n^\top$. The IBP is suitable to use as a prior process in latent feature models, since it generates possibly infinite number of columns and adaptively adjust the number of features on given dataset. Hence, with an IBP prior we need not specify the number of features in advance.

One interesting observation is that while it is a marginal of the beta-Bernoulli processes (Thibaux & Jordan, 2007), the IBP may also be understood as a limit of the finite-dimensional beta-Bernoulli process. More specifically, the IBP with parameter $\alpha > 0$ can be obtained as

$$\pi_k \sim \text{beta}(\alpha/K, 1), \quad z_{n,k} \sim \text{Ber}(\pi_k), \quad K \to \infty. \tag{7}$$

This beta-Bernoulli process naturally induces sparsity in the latent feature allocation matrix $\mathbf{Z}$. As $K \to \infty$, the expected number of nonzero entries in $\mathbf{Z}$ converges to $N\alpha$ (Griffiths & Ghahramani, 2005) , where $\alpha$ is a hyperparameter to control the overall sparsity level of $\mathbf{Z}$.

In this paper, we relate the latent feature models (6) to neural networks with dropout mask. Specifically, the binary random variables $z_{n,k}$ correspond to the dropout indicator, and the features $\mathbf{w}$ correspond to the inputs or intermediate units in neural networks. From this connection, we can think of a hierarchical Bayesian model where we place the IBP, or finite-dimensional beta-Bernoulli priors for the binary dropout indicators. We expect that due to the property of the IBP favoring sparse model, the resulting neural network would also be sparse.

### 3.3 Dependent Indian Buffet Processes

One important assumption in the IBP is that features are *exchangeable* - the distribution is invariant to the permutation of feature assignments. This assumption makes the posterior inference convenient, but restricts flexibility when we want to model the dependency of feature assignments to the input covariates $\mathbf{x}$, such as times or spatial locations. To this end, Williamson et al. Williamson et al. (2010) proposed dependent Indian Buffet processes (dIBP), which triggered a line of follow-up work (Zhou et al., 2011; Ren et al., 2011). These models can be summarized as following generative process:

$$\boldsymbol{\pi} \sim p(\boldsymbol{\pi}) \quad z|\boldsymbol{\pi}, \mathbf{x} \sim \text{Ber}(g(\boldsymbol{\pi}, \mathbf{x})), \tag{8}$$

where $g(\cdot, \cdot)$ is an arbitrary function that maps $\boldsymbol{\pi}$ and $\mathbf{x}$ to a probability. In our latent feature interpretation for neural network layers above, the input covariates $\mathbf{x}$ corresponds to the input or activations in the previous layer. In other words, we build a data-dependent dropout model where the dropout rates depend on the inputs. In the main contribution section, we will further explain how we will construct this data-dependent dropout layers in detail.

## 4 Main contribution

### 4.1 Variational Beta-Bernoulli dropout

Inspired by the latent-feature model interpretation of layers in neural networks, we propose a Bayesian neural network layer overlaid with binary random masks sampled from the finite-dimensional beta-Bernoulli prior. Specifically, let $\mathbf{W}$ be a parameter of a neural network layer, and let $\mathbf{z}_n \in \{0,1\}^K$ be a binary mask vector to be applied for the $n$-th observation $\mathbf{x}_n$. The dimension of $\mathbf{W}$ needs not be equal to $K$. Instead, we may enforce arbitrary group sparsity by sharing the binary masks among multiple elements of $\mathbf{W}$. For instance, let $\mathbf{W} \in \mathbb{R}^{K \times L \times M}$ be a parameter tensor in a convolutional neural network with $K$ channels. To enforce a channel-wise sparsity, we introduce $\mathbf{z}_n \in \{0,1\}^K$ of

$K$ dimension, and the resulting masked parameter $\widetilde{\mathbf{W}}_n$ for the $n$-th observation is given as

$$\{z_{n,k}W_{k,\ell,m} \,|\, (k,\ell,m) = (1,1,1), \ldots, (K,L,M)\}, \tag{9}$$

where $W_{k,\ell,m}$ is the $(k,\ell,m)$-th element of $\mathbf{W}$. From now on, with a slight abuse of notation, we denote this binary mask multiplication as

$$\widetilde{\mathbf{W}}_n = \mathbf{z}_n \otimes \mathbf{W}, \tag{10}$$

with appropriate sharing of binary mask random variables. The generative process of our Bayesian neural network is then described as

$$\mathbf{W} \sim \mathcal{N}(\mathbf{0}, \lambda \mathbf{I}), \ \ \boldsymbol{\pi} \sim \prod_{k=1}^{K} \text{beta}(\pi_k; \alpha/K, 1), \ \ \mathbf{z}_n|\boldsymbol{\pi} \sim \prod_{k=1}^{K} \text{Ber}(z_{n,k}; \pi_k), \ \ \widetilde{\mathbf{W}}_n = \mathbf{z}_n \otimes \mathbf{W}. \tag{11}$$

Note the difference between our model and the model in Gal & Ghahramani (2016). In Gal & Ghahramani (2016), only Gaussian prior is placed on the parameter $\mathbf{W}$, and the dropout is applied in the variational distribution $q(\mathbf{W})$ to approximate $p(\mathbf{W}|\mathcal{D})$. Our model, on the other hand, includes the binary mask $\mathbf{z}_n$ in the prior, and the posterior for the binary masks should also be approximated.

The goal of the posterior inference is to compute the posterior distribution $p(\mathbf{W}, \mathbf{Z}, \boldsymbol{\pi}|\mathcal{D})$, where $\mathbf{Z} = \{\mathbf{z}_1, \ldots, \mathbf{z}_N\}$. We approximate this posterior with the variational distribution of the form

$$q(\mathbf{W}, \mathbf{Z}, \boldsymbol{\pi}|\mathbf{X}) = \delta_{\widehat{\mathbf{W}}}(\mathbf{W}) \prod_{k=1}^{K} q(\pi_k) \prod_{n=1}^{N} \prod_{k=1}^{K} q(z_{n,k}|\pi_k), \tag{12}$$

where we omitted the indices of layers for simplicity. For $\mathbf{W}$, we conduct computationally efficient point-estimate to get the single value $\widehat{\mathbf{W}}$, with the weight-decay regularization arising from the zero-mean Gaussian prior. For $\boldsymbol{\pi}$, following Nalisnick & Smyth (2017), we use the Kumaraswamy distribution (Kumaraswamy, 1980) for $q(\pi_k)$:

$$q(\pi_k; a_k, b_k) = a_k b_k \pi_k^{a_k-1} (1 - \pi_k^{a_k})^{b_k-1}, \tag{13}$$

since it closely resembles the beta distribution and easily reparametrizable as

$$\pi_k(u; a_k, b_k) = (1 - u^{\frac{1}{b_k}})^{\frac{1}{a_k}}, \quad u \sim \text{unif}([0, 1]). \tag{14}$$

We further assume that $q(z_{n,k}|\pi_k) = p(z_{n,k}|\pi_k) = \text{Ber}(\pi_k)$. $z_k$ is sampled by reparametrization with continuous relaxation (Maddison et al., 2017; Jang et al., 2017; Gal et al., 2017),

$$z_k = \text{sgm}\left(\frac{1}{\tau}\left(\log \frac{\pi_k}{1 - \pi_k} + \log \frac{u}{1 - u}\right)\right), \tag{15}$$

where $\tau$ is a temperature of continuous relaxation, $u \sim \text{unif}([0, 1])$, and $\text{sgm}(x) = \frac{1}{1+e^{-x}}$. The KL-divergence between the prior and the variational distribution is then obtained in closed form as follows (Nalisnick & Smyth, 2017):

$$\text{D}_{\text{KL}}[q(\mathbf{Z}, \boldsymbol{\pi}) \| p(\mathbf{Z}, \boldsymbol{\pi})] = \sum_{k=1}^{K} \left\{ \frac{a_k - \alpha}{a_k}\left(-\gamma - \Psi(b_k) - \frac{1}{b_k}\right) + \log \frac{a_k b_k}{\alpha} - \frac{b_k - 1}{b_k} \right\}, \tag{16}$$

where $\gamma$ is Euler-Mascheroni constant and $\Psi(\cdot)$ is the digamma function. Note that the infinite series in the KL-divergence vanishes because of the choice $p(\pi_k) = \text{beta}(\pi_k; \alpha/K, 1)$.

We can apply the SGVB framework described in Section 3.1 to optimize the variational parameters $\{a_k, b_k\}_{k=1}^{K}$. After the training, the prediction for a novel input $\mathbf{x}_*$ is given as

$$p(\mathbf{y}_*|\mathbf{x}_*, \mathcal{D}, \mathbf{W}) = \mathbb{E}_{p(\mathbf{z}_*, \boldsymbol{\pi}, \mathbf{W}|\mathcal{D})}[p(\mathbf{y}_*|\mathbf{f}(\mathbf{x}_*; \mathbf{z}_* \otimes \mathbf{W}))] \approx \mathbb{E}_{q(\mathbf{z}_*, \boldsymbol{\pi})}[p(\mathbf{y}_*|\mathbf{f}(\mathbf{x}_*; \mathbf{z}_* \otimes \widehat{\mathbf{W}}))], \tag{17}$$

and we found that the following naïve approximation works well in practice,

$$p(\mathbf{y}_*|\mathbf{x}_*, \mathcal{D}, \mathbf{W}) \approx p(\mathbf{y}_*|\mathbf{f}(\mathbf{x}_*; \mathbb{E}_q[\mathbf{z}_*] \otimes \widehat{\mathbf{W}})), \tag{18}$$

where

$$\mathbb{E}_q[z_{*,k}] = \mathbb{E}_{q(\pi_k)}[\pi_k], \quad \mathbb{E}_{q(\pi_k)}[\pi_k] = \frac{b_k \Gamma(1 + a_k^{-1}) \Gamma(b_k)}{\Gamma(1 + a_k^{-1} + b_k)}. \tag{19}$$

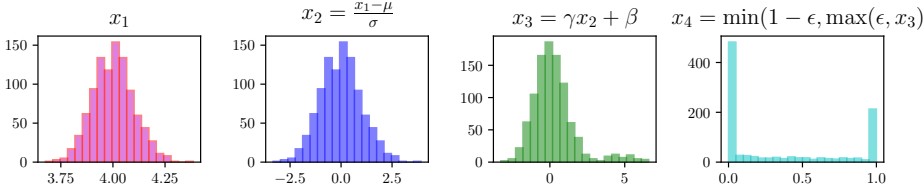

Figure 1: An example to show the intuition behind Eq (21). Each block represents a histogram of output distribution. A set of activations (1st block) are standardized (2nd block), and rescaled and shifted (3rd block), and transformed into probabilities (4th block).

## 4.2 VARIATIONAL DEPENDENT BETA-BERNOULLI DROPOUT

Now we describe our Bayesian neural network model with input dependent beta-Bernoulli dropout prior constructed as follows:

$$\mathbf{W} \sim \mathcal{N}(\mathbf{0}, \lambda \mathbf{I}), \quad \boldsymbol{\pi} \sim \prod_{k=1}^{K} \text{beta}(\pi_k; \alpha/K, 1), \quad \mathbf{z}_n | \boldsymbol{\pi}, \mathbf{x}_n \sim \prod_{k=1}^{K} \text{Ber}(z_{n,k}; \varphi_k(x_{n,k})), \quad (20)$$

Here, $\mathbf{x}_n$ is the input to the dropout layer. For convolutional layers, we apply the global average pooling to tensors to get vectorized inputs. In principle, we may introduce another fully connected layer as $(\varphi_1(x_{n,1}), \ldots, \varphi_K(x_{n,K})) = \text{sgm}(\mathbf{V}\mathbf{x}_n + \mathbf{c})$, with additional parameters $\mathbf{V} \in \mathbb{R}^{K \times K}$ and $\mathbf{c} \in \mathbb{R}^K$, but this is undesirable for the network sparsification. Rather than adding parameters for fully connected layer, we propose simple yet effective way to generate input-dependent probability, with minimal parameters involved. Specifically, we construct each $\pi_k(x_{n,k})$ independently as follows:

$$\varphi_k(x_{n,k}) = \pi_k \min \left( 1 - \epsilon, \max \left( \epsilon, \gamma_k \frac{x_{n,k} - \mu_k}{\sigma_k} + \beta_k \right) \right), \quad (21)$$

where $\mu_k$ and $\sigma_k$ are the estimates of $k$-th components of mean and standard deviation of inputs, and $\gamma_k$ and $\beta_k$ are scaling and shifting parameters to be learned, and $\epsilon > 0$ is some small tolerance to prevent overflow. The parameterization in (21) is motivated by the batch normalization (Ioffe & Szegedy, 2015). The intuition behind this construction is as follows. The inputs after the standardization would approximately be distributed as $\mathcal{N}(0, 1)$, and thus would be centered around zero. If we pass them through $\max(\epsilon, \min(1 - \epsilon))$, most of insignificant dimensions would have probability close to zero. However, some inputs may be important regardless of the significance of current activation. In that case, we expect the corresponding shifting parameter $\beta_k$ to be large. Thus by $\boldsymbol{\beta} = (\beta_1, \ldots, \beta_K)$ we control the overall sparsity, but we want them to be small unless required to get sparse outcomes. We enforce this by placing a prior distribution on $\boldsymbol{\beta} \sim \mathcal{N}(\mathbf{0}, \rho \mathbf{I})$.

The goal of variational inference is hence to learn the posterior distribution $p(\mathbf{W}, \mathbf{Z}, \boldsymbol{\pi}, \boldsymbol{\beta} | \mathcal{D})$, and we approximate this with variational distribution of the form

$$q(\mathbf{W}, \mathbf{Z}, \boldsymbol{\pi}, \boldsymbol{\beta} | \mathbf{X}) = \delta_{\widehat{\mathbf{W}}}(\mathbf{W}) \prod_{k=1}^{K} q(\pi_k) q(\beta_k) \prod_{n=1}^{N} \prod_{k=1}^{K} q(z_{n,k} | \pi_k, \mathbf{x}_n), \quad (22)$$

where $q(\pi_k)$ are the same as in beta-Bernoulli dropout, $q(\beta_k) = \mathcal{N}(\beta_k; \eta_k, \kappa_k^2)$, and $q(z_{n,k} | \pi_k) = p(z_{n,k} | \pi_k, \mathbf{x}_n)$ [1] The KL-divergence is computed as

$$D_{\text{KL}}[q(\mathbf{Z}, \boldsymbol{\pi} | \mathbf{X}) \| p(\mathbf{Z}, \boldsymbol{\pi})] + D_{\text{KL}}[q(\boldsymbol{\beta}) \| p(\boldsymbol{\beta})], \quad (23)$$

where the first term was described for beta-Bernoulli dropout and the second term can be computed analytically.

The prediction for the novel input $\mathbf{x}_*$ is similarity done as in the beta-Bernoulli dropout, with the näive approximation for the expectation:

$$p(\mathbf{y}_* | \mathbf{x}_*, \mathcal{D}, \mathbf{W}) \approx p(\mathbf{y}_* | \mathbf{f}(\mathbf{x}_*; \mathbb{E}_q[\mathbf{z}_*] \otimes \widehat{\mathbf{W}})), \quad (24)$$

---

[1] In principle, we may introduce an inference network $q(z | \pi, \mathbf{x}, \mathbf{y})$ and minimizes the KL-divergence between $q(z | \boldsymbol{\pi}, \mathbf{x}, \mathbf{y})$ and $p(\mathbf{z} | \boldsymbol{\pi}, \mathbf{x})$, but this results in discrepancy between training and testing for sampling $\mathbf{z}$, and also make optimization cumbersome. Hence, we chose to simply set them equal. Please refer to Sohn et al. (2015) for discussion about this.

Table 1: Results for LeNet-500-300 and LeNet5-Caffe on MNIST. Error and Memory are in %.

|  | LeNet 500-300 | | | | LeNet5-Caffe | | | |
|---|---|---|---|---|---|---|---|---|
|  | Error | Neurons | Speedup | Memory | Error | Neurons/Filters | Speedup | Memory |
| Original | 1.56 | 784-500-300 | 1.0 | 100.0 | 0.7 | 20-50-800-500 | 1.0 | 100.0 |
| SSL | $2.30 \pm 0.09$ | 404-32-22 | 39.13 | 2.55 | $0.97 \pm 0.06$ | 6-8-107-10 | **12.74** | **10.97** |
| SVD | $1.50 \pm 0.04$ | 532-64-35 | 14.73 | 6.78 | $0.72 \pm 0.01$ | 10-16-238-29 | 5.18 | 13.97 |
| SBP | $1.59 \pm 0.05$ | 255-100-43 | 17.91 | 5.57 | $0.74 \pm 0.02$ | 10-18-128-38 | 4.81 | 13.75 |
| BB | $\mathbf{1.34 \pm 0.04}$ | 294-110-71 | 13.26 | 7.52 | $\mathbf{0.57 \pm 0.01}$ | 13-25-156-54 | 2.94 | 16.2 |
| DBB | $1.38 \pm 0.07$ | 106-56-46 | **33.51** | **0.98** | $0.63 \pm 0.02$ | 13-24-53-27 | 2.95 | 14.79 |
| VIB | $1.48 \pm 0.07$ | 139-101-28 | 31.39 | 3.17 | $0.71 \pm 0.02$ | 12-18-82-34 | 4.02 | 14.35 |
| GD | $1.54 \pm 0.04$ | 488-142-136 | 6.04 | 16.54 | $0.66 \pm 0.03$ | 14-29-368-174 | 2.36 | 28.15 |

where

$$\mathbb{E}_q[z_{*,k}] = \mathbb{E}_q[\pi_k] \min\left(1 - \epsilon, \max\left(\epsilon, \gamma_k \frac{x_{n,k} - \mu_k}{\sigma_k} + \eta_k\right)\right). \quad (25)$$

**Two stage pruning scheme** Since $\pi_k \geq \pi_k(x_{n,k})$ for all $x_{n,k}$, we expect the resulting network to be sparser than the network pruned only with the beta-Bernoulli dropout (only with $\pi_k$). To achieve this, we propose a two-stage pruning scheme, where we first prune the network with beta-Bernoulli dropout, and prune the network again with $\pi_k(x_{n,k})$ while holding the variables $\boldsymbol{\pi}$ fixed. By fixing $\boldsymbol{\pi}$ the resulting network is guaranteed to be sparser than the network before the second pruning.

## 5 EXPERIMENTS

We now compare our beta-Bernoulli dropout (BB) and input-dependent beta-Bernoulli dropout (DBB) to other structure learning/pruning algorithms on several neural networks using benchmark datasets.

**Experiment Settings** We followed a common setting to compare pruning algorithms by using LeNet 500-300, LeNet 5-Caffe [2], and VGG-like (Zagoruyko, 2015) networks on MNIST LeCun et al. (1998), CIFAR-10, and CIFAR-100 datasets (Krizhevsky & Hinton, 2009). We included recent Bayesian pruning methods for a fair comparison: sparse variational dropout (SVD Molchanov et al. (2017)), structured sparsity learning (SSL Wen et al. (2016)) and structured Bayesian pruning (SBP Neklyudov et al. (2017)), variational information bottleneck (VIB Dai et al. (2018)), and generalized dropout (GD Srinivas & Babu (2016)). We faithfully tuned all hyperparameters of baselines on a validation set to find a reasonable solution that is well balanced between accuracy and sparsification, while fixing batch size (100) and the number of maximum epochs (200) to match our experiment setting.

**Implementation Details** We pretrained all networks using the standard training procedure before fine-tuning for network sparsification Molchanov et al. (2017); Neklyudov et al. (2017). While pruning, we set the learning rate for the weights $\mathbf{W}$ to be 0.1 times smaller than those for the variational parameters as in Neklyudov et al. (2017). We used Adam (Kingma & Ba, 2015) for all methods. For DBB, as mentioned in Section 4.2, we first prune networks with BB, and then prune again with DBB whiling holding the variational parameters for $q(\boldsymbol{\pi})$ fixed.

We report all hyperparameters of BB and DBB for reproducing our results. We set $\alpha/K = 10^{-4}$ for all layers of BB and DBB. In principle, we may fix $K$ to be large number and tune $\alpha$. However, in the network sparsification tasks, $K$ is given as the neurons/filters to be pruned. Hence, we chose to set the ratio $\alpha/K$ to be small number altogether. In the testing phase, we pruned the neurons/filters whose expected dropout mask probability $\mathbb{E}_q[\pi_k]$ are smaller than a fiixed threshold $10^{-33}$. For the input-dependent dropout, since the number of pruned neurons/filters differ according to the inputs, we report them as the running average over the test data. We fixed the temperature parameter of concrete distribution $\tau = 10^{-1}$ and the prior variance of $\boldsymbol{\beta}$, $\rho = \sqrt{5}$ for all experiments.

---

[2] https://github.com/BVLC/caffe/blob/master/examples/mnist
[3] We tried different values such as $10^{-2}$ or $10^{-4}$, but the difference was insignificant.

Table 2: Classification accuracy and sparsification performance of various pruning methods on CIFAR-10 and CIFAR-100 datasets. Error and Memory are in %.

| CIFAR-10 | | | | |
|---|---|---|---|---|
| | Error | Filters/Neurons | Speedup | Memory |
| Original | 7.43 | 64-64-128-128-256-256-256-512-512-512-512-512-512-512-512 | 1.00 | 100.0 |
| SSL | 8.27 ± 0.11 | 64-62-128-115-241-113-32-63-53-12-89-52-250-250-442 | 1.67 | 14.78 |
| SVD | 7.85 ± 0.08 | 64-64-128-128-254-209-54 -107-102-14-162-121-356-386-490 | 1.41 | 21.18 |
| SBP | 7.27 ± 0.06 | 64-64-128-128-254-209-55-107-100-14-155-125-98-92-353 | 1.41 | 18.546 |
| BB | **6.66 ± 0.11** | 64-64-128-126-251-174-41-89-81-13-118-108-23-23-68 | 1.49 | 16.09 |
| DBB | 7.03 ± 0.16 | 63 -61-126-122-236-131-36-64-65-11-93-93-17-18-48 | 1.54 | 14.08 |
| VIB | 7.17 ± 0.16 | 57-63-119-118-206-135-38-59-54-13-86-90-9-9-45 | **1.78** | **12.93** |
| GD | 7.26 ± 0.15 | 64-64-128-128-254-207-53-105-103-15-161-125-376-340-366 | 1.41 | 20.97 |

| CIFAR-100 | | | | |
|---|---|---|---|---|
| | Error | Filters | Speedup | Memory |
| Original | 31.46% | 64-64-128-128-256-256-256-512-512-512-512-512-512-512-512 | 1.0 | 100.0 |
| SSL | 32.75±0.16 | 62-64-128-128-255-254-135-187-98-22-212-156-512-512-512 | 1.28 | 28.24 |
| SVD | 31.26 ±0.06 | 64-64-128-128-255-255-135-238-173-26-343-268-512-512-512 | 1.25 | 36.77 |
| SBP | 30.56 ± 0.03 | 64-64-128-128-255-255-135-238-173-26-345-270-450-429-512 | 1.25 | 35.74 |
| BB | 29.27 ± 0.27 | 64-64-128-128-255-255-134-213-152-25-268-256-47-47-194 | 1.26 | 26.15 |
| DBB | **28.85 ± 0.06** | 63-63-126-126-245-234-129-167-124-22-223-227-35-36-104-105 | 1.29 | **22.64** |
| VIB | 29.87 ± 0.19 | 62-64-128-127-252-239-129-204-115-23-210-196-26-26-154 | **1.31** | 22.74 |
| GD | 30.54 ± 0.23 | 64-64-128-128-253-245-115-214-164-26-320-254-415-364-434 | 1.29 | 32.42 |

Figure 2: Parts of $\varphi(\mathbf{x})$ in 3rd, 8th, 15th layer of VGG network for CIFAR-100, w.r.t. different inputs.

## 5.1 EXPERIMENTS ON MNIST DATASET

We used LeNet 500-300 and LeNet 5-Caffe networks on MNIST for comparison. Following the conventions, we applied dropout to the inputs to the fully connected layers and right after the convolution for the convolutional layers. We report the results with basic settings in Table 1. Please refer to the appendix where we presented various results to highlight the tradeoff between sparsity and accuracy (Table 3). For both neworks, BB and DBB achieved significantly higher accuracy than ther methods. On LeNet-500-300, DBB pruned large amount of neurons in the input layer, because the inputs to this network are simply vectorized pixel values, so it can prune the inputs according to the digit classes (Fig. 3). Also, we found that the dropout masks generated by DBB tend to be generic at lower network layers to extract common features, but become class-specific at higher layers to specialize features for class discriminativity. See the appendix for the additional results on LeNet-5-Caffe showing this tendency (Fig. 3).

## 5.2 EXPERIMENTS ON CIFAR-10 AND CIFAR-100 DATASETS

We compared the pruning algorithms on VGG-like network adapted for CIFAR-10 and CIFAR-100 datasets. Table 2 summarizes the performance of each algorithm on particular setting, where BB and DBB achieved impressive sparsity with significantly improved accuracy. Further analysis of the filters retained by DBB in Fig. 2 shows that DBB either retains most filters (layer 3) or perform generic pruning (layer 8) at lower layers, while performing diversified pruning at higher layers (layer 15). Further, at layer 15, instances from the same class retained similar filters, while instances from different classes retained different filters. Please refer to the appendix, where we presented tradeoff between accuracy and sparsity with various settings (Table 4).

## 6 CONCLUSION

We have proposed novel beta-Bernoulli dropout for network regularization and sparsification, where we learn dropout probabilities for each neuron either in an input-independent or input-dependent

manner. Our beta-Bernoulli dropout learns the distribution of sparse Bernoulli dropout mask for each neuron in a variational inference framework, in contrast to existing work that learned the distribution of Gaussian multiplicative noise or weights, and obtains significantly more compact network compared to those competing approaches. Further, our dependent beta-Bernoulli dropout that input-adaptively decides which neuron to drop further improves on the input-independent beta-Bernoulli dropout, both in terms of size of the final network obtained and run-time computations.

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

# Appendices

## A   SOME QUALITATIVE RESULTS FOR DBB ON MNIST EXPERIMENTS

(Fig. 3, left) shows the class average values of $\varphi(\mathbf{x})$ for the class 0, 3, 7, and 9. Unlike the other pruning algorithms where the general background is pruned, DBB specifically pruned the area where the digits present. (Fig. 3, right) shows the correlation coefficients between the class average values of $\varphi(\mathbf{x})$ learnt from DBB. As we stated in the paper, it clearly show the tendency to share filters in lower layers, and be specific in higher layers. This tendency can also be observed in the experiments with VGG on CIFAR10 and CIFAR100.

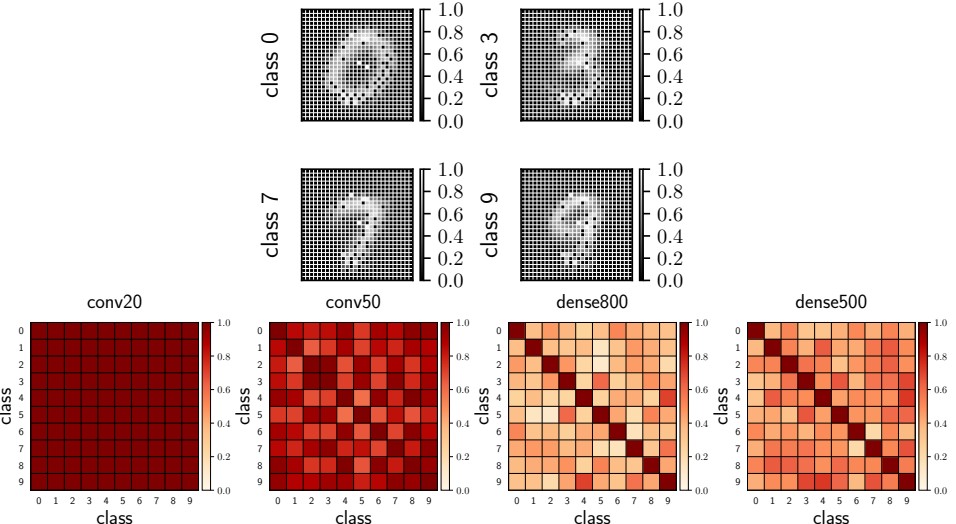

Figure 3: Correlation coefficients of class averages of $\varphi(\mathbf{x})$ for the four layers in LeNet5-Caffe.

## B   ADDITIONAL RESULTS ON LENET-500-300 AND LENET5-CAFFE

Table 3 shows the results of pruning algorithms with five sparsity levels. We controlled the sparsity by tuning the initial learning rate for sparsity related variables (for instance, the learning rate for the paramters of $q(\pi_k; a_k, b_k)$ for BB). For SSL, we tested five sparsity rate parameter $\{2 \cdot 10^{-3}, 10^{-3}, 5 \cdot 10^{-4}, 10^{-4}, 5 \cdot 10^{-5}\}$, and for all the other methods we tested with initial learning rate $\{5 \cdot 10^{-2}, 2 \cdot 10^{-2}, 10^{-2}, 5 \cdot 10^{-3}, 10^{-3}\}$. The results presented in the main text are the ones with sparsity rate $5 \cdot 10^{-4}$ for SSL and initial learning rate $10^{-2}$ for all other methods. We ran all the algorithms five times and reported mean and standard deviations.

## C   ADDITIONAL RESULTS ON VGG-LIKE

Table 4 shows the results of pruning algorithms with four sparsity levels. We controlled the sparsity by tuning the initial learning rate for sparsity related variables. For SSL, we tested with sparsity rate parameter $\{10^{-3}, 5 \cdot 10^{-4}, 10^{-4}, 5 \cdot 10^{-5}\}$, and for all the other methods we tested with initial learning rate $\{2 \cdot 10^{-2}, 10^{-2}, 5 \cdot 10^{-3}, 10^{-3}\}$. The results presented in the main text are the ones with sparsity rate $5 \cdot 10^{-4}$ for SSL and initial learning rate $10^{-2}$ for all other methods. We ran all three algorithms five times and reported mean and standard deviations.

Table 3: Comparision of pruning methods on LeNet-500-300 and LeNet5-Caffe with MNIST. Error and Memory are in %.

| | LeNet 500-300 | | | LeNet5-Caffe | | |
|---|---|---|---|---|---|---|
| | Error | Speedup | Memory | Error | Speedup | Memory |
| SSL | $3.52 \pm 0.12$ | 75.69 | 1.32 | $1.44 \pm 0.08$ | 23.17 | 10.09 |
| | $2.63 \pm 0.12$ | 54.83 | 1.82 | $1.03 \pm 0.05$ | 20.26 | 10.20 |
| | $2.30 \pm 0.09$ | 39.13 | 2.55 | $0.97 \pm 0.06$ | 12.74 | 10.97 |
| | $2.06 \pm 0.13$ | 27.68 | 3.61 | $0.84 \pm 0.05$ | 8.06 | 11.83 |
| | $1.93 \pm 0.09$ | 22.99 | 4.34 | $0.83 \pm 0.07$ | 6.77 | 12.30 |
| SVD | $2.75 \pm 0.14$ | 21.44 | 4.66 | $1.47 \pm 0.11$ | 3.62 | 16.38 |
| | $1.51 \pm 0.04$ | 18.39 | 5.43 | $0.74 \pm 0.04$ | 4.92 | 14.05 |
| | $1.50 \pm 0.04$ | 14.73 | 6.78 | $0.72 \pm 0.01$ | 5.17 | 14.04 |
| | $1.51 \pm 0.03$ | 11.99 | 8.33 | $0.71 \pm 0.01$ | 4.78 | 14.75 |
| | $1.45 \pm 0.02$ | 8.40 | 11.89 | $0.68 \pm 0.02$ | 2.92 | 17.95 |
| SBP | $1.77 \pm 0.04$ | 24.81 | 4.01 | $1.06 \pm 0.27$ | 26.10 | 10.54 |
| | $1.70 \pm 0.05$ | 21.60 | 4.61 | $0.79 \pm 0.05$ | 4.81 | 13.51 |
| | $1.59 \pm 0.05$ | 17.91 | 5.57 | $0.74 \pm 0.02$ | 4.81 | 13.75 |
| | $1.52 \pm 0.03$ | 14.76 | 6.75 | $0.76 \pm 0.03$ | 4.06 | 14.64 |
| | $1.51 \pm 0.01$ | 11.46 | 8.70 | $0.73 \pm 0.02$ | 2.89 | 16.94 |
| BB | $1.60 \pm 0.09$ | 30.35 | 3.28 | $0.65 \pm 0.03$ | 3.45 | 14.75 |
| | $1.38 \pm 0.04$ | 17.24 | 5.78 | $0.63 \pm 0.03$ | 3.38 | 14.94 |
| | $1.34 \pm 0.04$ | 13.26 | 7.52 | $0.57 \pm 0.01$ | 2.94 | 16.2 |
| | $1.34 \pm 0.02$ | 10.78 | 9.25 | $0.58 \pm 0.04$ | 2.72 | 17.43 |
| | $1.26 \pm 0.03$ | 5.08 | 19.68 | $0.63 \pm 0.01$ | 2.43 | 36.23 |
| DBB | $2.01 \pm 0.07$ | 64.88 | 0.50 | $0.77 \pm 0.03$ | 4.11 | 12.75 |
| | $1.50 \pm 0.09$ | 43.24 | 0.69 | $0.64 \pm 0.02$ | 3.61 | 13.52 |
| | $1.38 \pm 0.07$ | 33.51 | 0.98 | $0.63 \pm 0.02$ | 2.95 | 14.79 |
| | $1.32 \pm 0.03$ | 23.47 | 2.08 | $0.60 \pm 0.04$ | 2.8 | 15.21 |
| | $1.26 \pm 0.02$ | 5.97 | 14.28 | $0.58 \pm 0.01$ | 2.45 | 26.68 |
| VIB | $2.06 \pm 0.07$ | 54.50 | 1.82 | $0.82 \pm 0.07$ | 4.52 | 13.74 |
| | $1.75 \pm 0.08$ | 35.80 | 2.78 | $0.77 \pm 0.05$ | 4.17 | 14.00 |
| | $1.48 \pm 0.07$ | 31.39 | 3.17 | $0.76 \pm 0.02$ | 4.02 | 14.17 |
| | $1.51 \pm 0.09$ | 28.57 | 3.48 | $0.71 \pm 0.04$ | 3.87 | 14.35 |
| | $1.48 \pm 0.03$ | 20.33 | 4.90 | $0.65 \pm 0.04$ | 3.23 | 15.31 |
| GD | $1.66 \pm 0.07$ | 10.68 | 9.34 | $0.73 \pm 0.03$ | 2.71 | 18.46 |
| | $1.53 \pm 0.05$ | 8.02 | 12.44 | $0.71 \pm 0.06$ | 2.60 | 22.63 |
| | $1.54 \pm 0.04$ | 6.04 | 16.54 | $0.66 \pm 0.03$ | 2.36 | 28.15 |
| | $1.50 \pm 0.02$ | 4.37 | 22.84 | $0.67 \pm 0.01$ | 2.15 | 35.91 |
| | $1.31 \pm 0.05$ | 3.18 | 31.42 | $0.60 \pm 0.02$ | 1.92 | 65.78 |

Table 4: Comparision of pruning methods on VGG-like with CIFAR10 and CIFAR100. Error and Memory are in %.

| | VGG-CIFAR10 | | | VGG-CIFAR100 | | |
|---|---|---|---|---|---|---|
| | Error | Speedup | Memory | Error | Speedup | Memory |
| SSL | $9.56 \pm 0.09$ | 2.09 | 12.01 | $36.27 \pm 0.22$ | 1.43 | 23.21 |
| | $8.27 \pm 0.11$ | 1.67 | 14.78 | $32.75 \pm 0.16$ | 1.28 | 28.24 |
| | $7.32 \pm 0.07$ | 1.42 | 19.04 | $30.64 \pm 0.06$ | 1.25 | 34.44 |
| | $7.12 \pm 0.16$ | 1.41 | 20.30 | $30.45 \pm 0.19$ | 1.25 | 36.14 |
| SVD | $9.71 \pm 0.18$ | 1.41 | 18.48 | $34.26 \pm 0.19$ | 1.25 | 35.76 |
| | $7.85 \pm 0.08$ | 1.41 | 21.18 | $31.26 \pm 0.06$ | 1.25 | 36.77 |
| | $7.78 \pm 0.01$ | 1.40 | 21.71 | $31.31 \pm 0.08$ | 1.25 | 36.89 |
| | $7.38 \pm 0.05$ | 1.40 | 22.47 | $31.22 \pm 0.06$ | 1.25 | 36.80 |
| SBP | $14.68 \pm 0.10$ | 1.43 | 17.72 | $45.03 \pm 1.59$ | 1.27 | 31.01 |
| | $7.27 \pm 0.06$ | 1.41 | 18.55 | $30.56 \pm 0.03$ | 1.25 | 35.74 |
| | $7.40 \pm 0.02$ | 1.40 | 21.03 | $30.82 \pm 0.02$ | 1.25 | 36.84 |
| | $7.40 \pm 0.02$ | 1.40 | 22.54 | $31.03 \pm 0.09$ | 1.25 | 36.90 |
| BB | $7.43 \pm 0.11$ | 1.89 | 12.45 | $40.13 \pm 1.84$ | 2.40 | 11.26 |
| | $6.66 \pm 0.11$ | 1.49 | 16.09 | $29.27 \pm 0.27$ | 1.26 | 26.15 |
| | $6.61 \pm 0.14$ | 1.42 | 17.79 | $28.89 \pm 0.07$ | 1.25 | 29.15 |
| | $6.51 \pm 0.06$ | 1.40 | 22.51 | $29.68 \pm 0.08$ | 1.25 | 36.90 |
| DBB | $8.89 \pm 0.12$ | 2.36 | 8.62 | $40.45 \pm 1.74$ | 2.63 | 9.39 |
| | $7.03 \pm 0.16$ | 1.54 | 14.08 | $28.85 \pm 0.06$ | 1.29 | 22.64 |
| | $6.59 \pm 0.07$ | 1.44 | 6.64 | $28.48 \pm 0.18$ | 1.27 | 26.93 |
| | $6.39 \pm 0.04$ | 1.42 | 22.02 | $28.74 \pm 0.10$ | 1.26 | 36.41 |
| VIB | $7.87 \pm 0.13$ | 2.32 | 10.42 | $31.64 \pm 0.33$ | 1.65 | 17.84 |
| | $7.17 \pm 0.06$ | 1.78 | 12.93 | $29.87 \pm 0.19$ | 1.31 | 22.74 |
| | $6.93 \pm 0.13$ | 1.45 | 16.51 | $29.90 \pm 0.34$ | 1.25 | 27.26 |
| | $7.11 \pm 0.08$ | 1.41 | 17.77 | $30.20 \pm 0.18$ | 1.25 | 31.69 |
| GD | $8.06 \pm 0.12$ | 1.58 | 17.84 | $36.35 \pm 0.50$ | 1.56 | 23.93 |
| | $7.26 \pm 0.15$ | 1.41 | 20.97 | $30.54 \pm 0.23$ | 1.29 | 34.42 |
| | $6.98 \pm 0.12$ | 1.41 | 21.55 | $29.94 \pm 0.10$ | 1.25 | 36.34 |
| | $6.59 \pm 0.09$ | 1.40 | 22.54 | $29.29 \pm 0.01$ | 1.25 | 36.90 |

