# OpenReview forum: "ADAPTIVE NETWORK SPARSIFICATION VIA DEPENDENT VARIATIONAL BETA-BERNOULLI DROPOUT"
_ICLR.cc/2019/Conference_

### Official Review · AnonReviewer3 · 2018-10-31
**Novel idea to improve compression with data-dependent structured dropout. Redundant references to IBP. Missing some experiments with start-of-the-art bayesian compression methods.**

**Rating:** 7
**Confidence:** 4

**Review:**

Summary
------------------

The authors propose a new method to sparsify DNNs based on a dropout induced by a Beta-Bernoulli prior. They further propose a data-dependent dropout by linking the Beta-Bernoulli prevalence to the inputs, achieving a higher sparsification rate. In the experimental section they show that the proposed method achieves better compression rates than other methods in the literature. However, experiments against some recent methods are missing. Also, some additional experiments using data-dependent dropouts not based on the Beta-Bernoulli prior would help to better disentangle the effects of the two contributions of the paper. Overall, the paper is well-written but the mentioning of the IBP is confusing. The authors devote quite a bit of space to the IBP when it is actually not used at all.

 Detailed comments
-------------------------

1)	Introduction

The paper is well motivated and the introduction of the paper clearly states the two main contributions of the paper: a Beta-Bernoulli dropout prior and a dependent Beta Bernoulli dropout prior.

2)	Background

Section 3.1 is a nice summary of variational inference for BNNs. On the other hand, Section 3.2 is misleading. The authors use this section to introduce the IBP process (a generative sequential process to generate samples from a random measure called the Beta-Bernoulli process). However, this is not used in the paper at all. Then they introduce the Beta-Bernoulli prior as a finite Beta-Bernoulli process. I find this quite convoluted. I would suggest to introduce the Beta-Bernoulli distribution as a prior directly, and state that for alpha/K this is a sparse-inducing prior (where the average number of features is given by \frac{\alpha}{1 + \frac{\alpha}{K} ). No need to mention the IBP or the Beta Bernoulli process.

3)	Main Contribution

I think the design of a link function that allows to implement a data-dependent Beta-Bernoulli dropout is one of the keys of the paper and I would suggest that the author clearly state this contribution at the beginning of the paper. I would also like to see the application of this link-function to other sparsity inducing priors different than the Beta-Bernoulli. This would allow to further understand the data-dependent contribution to the final performance and how transferable this is to other settings. Also, Have the authors try to train the data-dependent Beta-Bernoulli from scratch, i.e. without the two steps approach? I am assuming the performance is worse, but I would publish the results for completeness.

4)	Experiments

The main issues with the experimental section are:
a)	I am missing some recent methods (some of them even cited in the related work section): e.g. Louizos et al. (2017). I would be interested in comparisons against the horshoe-prior and a data-dependent version of it. Also, a recent paper based on the variational information bottleneck have been recently published outperforming the state of the art in the field (http://proceedings.mlr.press/v80/dai18d.html).
b)	Table 1 should report the variance or other uncertainty measure: Given that they run the experiments 5 times, I do not understand why they only report the median. I would encourage the authors to publish the mean and the variance (at least).
In addition, one of my main question about the method is, once the network has been sparsified, how does this translate into a real performance improvement (in terms of memory and speed). In term of memory, you can always apply a standard compression algorithm. If the sparsity is about a certain threshold, you can resort to sparse-matrix implementations. However, regarding the speed only when you reach a certain sparsity level you would get a tangible improvement if your DL framework support sparse matrices. However, if you get an sparsity level below this threshold, e.g. 20%, you cannot resort to sparse matrices and therefore you would not get a speed improvement, unless you enforce structure sparsity or you optimize to low-level matrix multiplication routines. Are the Speedup/Memory results reported in Table 1 real or theoretical?

---

> ### Author Response · Authors · 2018-11-25
> **Response to Reviewer 3**
>
> - Misleading section 3.2.
> While our work was first motivated by IBP, we agree on your point that introducing IBP might not be necessary. We will restructure the paper such that we first introduce beta-Bernoulli process, and then briefly highlight its relationship to the IBP as an infinite limit.
>
> - Main contribution.
> Thanks for your suggestion. We totally agree on your point that our data dependent beta-Bernoulli (DBB) can be applied to other sparsity inducing priors. As you mentioned, we can train DBB from scratch, while generally it produces less accurate results with similar sparsity.
>
> - Missing several recent works.
> We implemented VIBNet (Dai et al., 2018) and included the results. Please refer to the updated paper. We also changed medians to mean +/ standard deviations for all tables. Currently the speedup/memory savings are theoretical values. As you mentioned, the real speedups or memory usages depend on various factors such as DL framework or hardwares.

---

### Official Review · AnonReviewer1 · 2018-11-02
**Interesting paper that needs more work**

**Rating:** 5
**Confidence:** 4

**Review:**

This work proposes Variational Beta-Bernoulli Dropout, a Bayesian way to sparsify neural networks by adopting Spike and Slab priors over the parameters of the network. Motivated by the Indian Buffet Process the authors further adopt Beta hyperpriors for the parameters of the Bernoulli distribution and also propose a way to set up the model such that it allows for input specific priors over the Bernoulli distributions. They then provide the necessary details for their variational approximations to the posterior distributions of both such models and experimentally validate their performance on the tasks of MNIST and CIFAR 10/100 classification.

This work is in general well written and conveys the main ideas in an clear manner. Furthermore, parametrising conditional group sparsity in a Bayesian way is also an interesting venue for research that can further facilitate for computational speedups for neural networks. The overall method seems simple to implement and doesn’t introduce too many extra learnable parameters.

Nevertheless, I believe that this paper needs more work in order to be published. More specifically:

- I believe that the authors need to further elaborate and compare with “Generalized Dropout”; the prior imposed on the weights for the non-dependent case is essentially the same with only small differences in the approximate posterior. Both methods seem to optimise, rather than integrate over, the weights of the network and the main difference is in how to handle the approximate distributions over the gates. Why would one prefer one parametrisation rather than the other? Furthermore, the authors of this work argue that they employ asymptotically unbiased gradients for the binary random variables, which is incorrect as the continuous relaxation provides a biased gradient estimator for the underlying discrete model.

- At section 3.2 the authors argue about the inherent sparsity inducing nature of the IBP model. In the finite K scenario this is not entirely the case as sparsity is only encouraged for alpha < K.

- At Eq. 11 the index “n” doesn’t make sense as the Bernoulli probability for each point depends only on the global pi_k. Similarly for Eq. 12.

- Since you tie q(z_nk|pi_k) = p(z_nk|pi_k) then it makes sense to phrase Eq.16 as just D_KL(q(pi) || p(pi)). Furthermore, I believe that you should properly motivate on why tying these two is a sensible thing to do.

- Figure 1 is misleading; you start from a unimodal distribution and then you simply apply a scalar scale and shift to the elements of that distribution. The output of that will always be a unimodal distribution but somehow you end up with a multimodal distribution on the third part of the figure. As a result, I believe that in this case you will not have two clear modes (one at 0 and one at 1) when you apply the hard-sigmoid rectification.

- The motivation for 21 seems a bit confusing to me; what do you mean with insignificant dimensions? What overflow does the epsilon prevent? If the input to the hard sigmoid is a N(0, 1) distribution then you will approximately have 1/3 of the activations having probability close to 1. Furthermore, it seems that you want beta to be small / negative to get sparse outcomes but the text implies that you want it to be large.

- It would be better to rewrite eq. 22 to include also the fact that you have a separate z per layer as currently it seems that the there is only one z. Furthermore, you have written that the variational posterior distribution depends on x_n on the RHS but not on the LHS.

- Above eq. 23 seems that it should be q(z_nk| pi_k, xn) = p(z_nk| pi_k, xn) rather than q(z_nk| pi_k) = p(z_nk| pi_k, xn)


Regarding the experiments; the MNIST results are not particularly convincing as the numbers are, in general, similar to other methods. Furthermore, Figure 2 is a bit small and confusing to read. Should FLOPS be on the y-axis or something else? Almost zero flops for the original model doesn’t seem right. Finally, at the CIFAR 10/100 experiment it seems that both BB and DBB achieve the best performance. However, it seems that the accuracy /sparsity obtained for the baselines is inferior to the results obtained on each of the respective papers. For example, SBP managed to get a 2.71x speedup with the VGG on CIFAR 10 and an error of 7.5%, whereas here the error was 8.68% with just 1.34x speedup. The extra visualisations provided at Figure 3 do look interesting though as it shows what the sparsity patterns learn.

---

> ### Author Response · Authors · 2018-11-25
> **Response to Reviewer 1**
>
> - Regarding generalized dropout:
> Thanks for pointing out that the continuous relaxation of binary latent variable gives biased gradient estimates. We stated that we are using 'asymptotically unbiased' gradient estimator, since the continuous relaxation becomes unbiased as the temperature parameter goes to zero. We also conducted series of experiments with generalized dropout and updated the revision. As you pointed out, beta-Bernoulli dropout (BB) is similar in that beta prior is placed on the dropout probabilities, but BB uses different gradient estimator and the range of hyperparameters for beta prior is different.
>
> - Condition alpha < K:
> As you mentioned, the sparsity of IBP is valid for infinite K, but also holds for finite but large K (alpha << K). As we mentioned in the main text, we fixed alpha/K = 1e-4 for all experiments (we controlled alpha according to K (number of neurons/filters)).
>
> - Index "n" in Eq.11 doesn't make sense.
> Even though we have global pi_k, we sample local z_n for each data point (i.e., z_1, z_2, … z_n ~(iid) Bern(pi)). This is related to the local reparametrization trick [Kingma et al. 2015], and reduces the variance of the gradient estimator.
>
> - Tying q(z_nk|pi_nk)=p(z_nk|pi_k).
> We explained the motivation for this choice in footnote 1 of page 6 in the paper. The main reason is to keep consistency between training and testing phase, since training is done with q(z_nk|pi_nk) and testing is done with p(z_nk|pi_k).
>
> - Figure 1 is misleading
> Each block of figure one represents a histogram of set of activation values. In third block, a bias vector beta is added, and due to the prior placed on beta, only small number of dimensions in beta has large value. Hence, the result of adding beta to the set of activations yields a bimodal distribution as in the third block of Figure 1.
>
> - Motivation for equation 21
> By insignificant dimensions we mean the dimensions whose activations are negative or close to zero, so the result of hard sigmoid function would result in mask probabilities close to zero. The epsilon prevents the overflow in the computation of logits, log (p/(1-p)). As we mentioned above, for beta, we want only small number of dimensions to be large, so that the result of adding beta would produce only small number of hard-sigmoid become close to one.
>
> - Better to rewrite eq. 22, 23
> Thanks for your suggestion. We have updated the draft based on your comments.
>
> - Figure 1 and 2 are difficult to interpret
> In the appendix, we provided all the figures in table instead of figures. Please refer to the revised paper.
>
> - Baseline results are not consistent with the reported ones
> We re-implemented all the baselines by ourselves and trained them under unified settings. The result of pruning algorithms depends heavily on the hyperparameters such as learning rate, batch size, number of iterations. We faithfully tuned all the hyperparameters of baseline algorithms and reported the best results. For instance, the codes for SBP released by the authors uses objective function \sum_{i=1}^n ELBO(x_i), while our code uniformly uses the objective function (\sum_{i=1}^n ELBO(x_i))/n, and this often makes big difference. We will release all the codes used to produce our experimental results upon acceptance of our paper.
>
> [Kingma et al. 15] Variational Dropout and the Local Reparametrization Trick, NIPS 2015

---

> > ### Comment · AnonReviewer1 · 2018-11-27
> > **Thank you for the clarifications. The paper is in a better state now but I still have my concerns.**
> >
> > Thank you for addressing my comments and for putting in the effort to revise the submission.
> >
> > - Regarding the Generalized Dropout / relaxation: Indeed the continuous relaxation becomes equivalent to the Bernoulli distribution when the temperature -> 0. However I wouldn't call it a gradient estimator in this case, as the derivative is zero everywhere except one point where it is infinite. Furthermore, while I appreciate the experimental comparison with Generalized Dropout, I would have liked a comparison / discussion on a more theoretical level that presents the advantages and drawbacks of each approximate posterior / prior hyperparameters.
> >
> > - Regarding the tying of q with p: While the tying of the input dependent prior and posterior over z was explained via the footnote, the explanation for the global case (i.e. not input dependent) was missing. It will be better to refer to the footnote in both cases, if the explanation is the same. Furthermore, why is testing done via p(z_nk | pi_k) and not q(z_nk| pi_k)? Usually, in order to obtain the predictive distribution we average over the posterior rather than the prior.
> >
> > - Regarding Figure 1: It makes sense that this can happen when \beta is a vector. From the captions above the figure, \beta was not bold typed so I assumed that it was a scalar. It would be better if you clarify this in text in order to avoid future misconceptions.
> >
> > - Regarding the results from SBP: I don't see how summing vs averaging can have such a difference on the results; it just changes the effective learning rate. In this case, you should have been able to replicate the same results if you just increased the learning rate by a factor of N.
> >
> > Overall, I believe that the submission is in a better shape now. Nevertheless, I am still not sure if it is good enough for a publication so I will not change my original score.

---

> > > ### Author Response · Authors · 2018-11-30
> > > **Answer to the concerns**
> > >
> > > Thanks for your comment.
> > >
> > > Comparison to generalized dropout
> > > Generalized dropout is similar to our beta-Bernoulli dropout in a sense that it places a beta prior Beta(alpha, beta) on the mask probability pi. The generalized dropout has several variants according to the choice of hyperparameters alpha and beta, and does not always lead to sparsity promoting algorithms. To get sparsity promoting effect, one may choose Dropout++(0) where alpha > 1 and beta = 1, or SAL where alpha < 1 and beta < 1. Beta-Bernoulli dropout does not correspond to any of these cases since we set alpha << 1 and beta = 1. The more important difference is in the learning procedure. Generalized dropout computes the point estimate of the mask probability pi, and optimize it via heuristics. In beta-Bernoulli dropout, we estimate an approximate posterior distribution of pi q(pi) instead of point estimates, with theoretically grounded concrete-Bernoulli gradient approximation, and these result in more robust results.
> > >
> > > Tying of q and p, why use p in testing?
> > > We wil try to clarify the tying of q(z_k|pi_k) and p(z_k|pi_k), and clarify that KL divergence between them vanishes. The reason for this choice is similar to the input dependent ones.
> > >
> > > Regarding the testing, let x_* be a test instance and (X, Y) be a training set. We have
> > > p(x_*| X, Y) = \int p(x_* | W, z_*) p(z_*|pi) p(W|X, Y) p(pi | X, Y) dpi dW dz_*,
> > > and p(W|X, Y), p(pi|X, Y) are approximated with learned q(W), q(pi). However, the binary mask z_* does not depend on the training set (X, Y), so the sampling should be done with p(z_*|pi). For training things are different because once we have observed labels Y the mask z depends on those labels, so the mask sampling should be done with true posterior p(z|pi, X, Y) approximated with q(z|pi). In our case, since we tie p and q, we use the same sampling distribution in training/testing.
> > >
> > > SBP implementation issue
> > > We have observed that the performance of SBP is sensitive to kl-scaling, initial learning rate from the issue of summing or averaging ELBOs, and the effect of batch normalization in the phase of fine-tuning. First off, we have found that the official code was implemented to employ kl-scaling in a factor of two. For a fair comparison, we didn’t use any kl-scaling for all baseline methods. Second, it is very critical to select an appropriate initial learning rate for achieving the reasonable balance between sparsity and accuracy, because the averaging ELBOs yields a different set of appropriate learning rates. Finally, we have observed that batch normalization plays a crucial role in achieving the competitive performance of SBP. When batch normalization always performs as a test mode in fine-tuning the network (i.e. using the statistics obtained in the pretraining phase), the performance of SBP is quite increased. We have found this interesting behavior of SBP in our code and the official code released by the authors. Now, we double-checked the code and tried our best to reproduce the results of all baseline methods including SBP.

---

### Official Review · AnonReviewer2 · 2018-11-04
**Confusion about inference**

**Rating:** 5
**Confidence:** 4

**Review:**

The authors propose a dropout method that uses the beta-Bernoulli process to learn the sparsity rate for each node.

The model itself make sense to me, though I don't have an understanding of why learning a node-specific sparsity rate should improve dropout -- i.e., what is there to learn? From what I understand about dropout, it's a stochastic method that has the same marginal as the original model, but because of the randomness induced it avoids bad local optimal solutions. Thus it's a learning trick, not a modeling technique. This treats dropout as something to directly model.

My confusion is mainly about inference. While there are many approximations introduced to make it work, if the sparsity z is something to be learned then why is it only being sampled from the beta prior in (15)? There is a likelihood term that incorporates z as well and it seems like this should be included as well to be strictly correct from a modeling standpoint. I didn't see any explanation in the discussion.

---

> ### Author Response · Authors · 2018-11-25
> **Response to Reviewer 2**
>
> - I don't have an understanding of why learning a node-specific dropout rate should improve dropout.
> Learning of node-specific dropout rate has been explored in several previous work, and it has been consistently reported that they improve generalization performance of deep neural networks over random ones. To list a few, [Kingma et al. 15] proposed to learn the variance of the multiplicative noise of Gaussian dropout, and [Gal et al. 17] proposed to learn the Bernoulli dropout rate by approximating it with concrete distribution.
>
> Our model is Bayesian and is a way to convert regular deep neural networks into Bayesian neural networks by approximating the optimal Bernoulli "noise" to the neurons. It is obvious that learning the distribution of the noise is a better approximation than setting it to some arbitrary distribution and hope that they coincide. Further, learning of per-neuron dropout rate means that we approximate the true distribution of an unknown noise distribution with a more accurate distribution compared to the use of a single distribution across all neurons.
>
> Another, more intuitive way to understand the generalization improvement with learned dropout is that injecting noise to neurons, if we propagate the noise to the input, is the same as injecting noise to inputs. Thus, dropout can be viewed as a data augmentation process to simulate inputs that may arrive at test time. Here, learning noise rate differently per neuron allows us to generate more relevant perturbations than random perturbations.
>
> - While there are many approximations introduced to make it work, if the sparsity z is something to be learned then why is it only being sampled from the beta prior in (15)?
> Please note that the goal of Eq (12) and below (including Eq (15)) is to define our variational distribution in order to approximate intractable true posterior. Under the variational inference framework, both prior Eq (11) and likelihood play crucial roles (this is the standard principal of variational inference; see Eq (3)). Overall, the sparsity inducing beta prior acts as a regularizer so that the model will find a configuration where the number of neurons activated is minimized (minimizing KL[q||p]) while maintaining the accuracy from the likelihood perspective (maximizing log p(X|theta)).
>
> [Kingma et al. 15] Variational Dropout and the Local Reparametrization Trick, NIPS 2015
> [Gal et al. 17] Concrete Dropout, NIPS 2017

---

### Author Response · Authors · 2018-11-25
**Summary of the updates**

Dear reviewers,

Thank you for your valuable comments and sorry for the late response. We summarized the updates in the revision below.

Main updates
- To address the comment of Reviewer 1 regarding the relation to generalized dropout [Srinivas and Babu, 2016] and the comment of Reviewer 3 regarding the comparison to the recent method, we implemented generalized dropout and VIBNet [Dai et al, 2018] and included the results in the main paper and appendix.

- Considering the comment of Reviewer 1 saying that the figures for LeNet experiments are hard to interpret, we replaced the plots with tables. We also presented the tradeoff results for VGG-like net on CIFAR10 and CIFAR100. In the main text, the representative results with basic sparsity settings are presented with actual number of neurons/filters learned. In the appendix, we presented the results with various sparsity levels, to highlight the tradeoff between sparsity and accuracy the pruning methods. Our general observation is that BB and DBB achieve highest accuracy given similar sparsity levels.

Minor edits

- According to the suggestion of Reviewer 3, we updated the errors to be reported with mean and standard deviations. The number of neurons/filters remaining, speedup in FLOPs, runtime memory savings are still reported in median.
- We edited equation (12), (22) and (23) according to the suggestion of Reviewer 1.

---

### Meta-Review · Area_Chair1 · 2018-12-14

**Confidence:** 4
**Recommendation:** Reject

**Metareview:**

The paper proposes Variational Beta-Bernoulli Dropout,, a Bayesian method for sparsifying neural networks. The method adopts a spike-and-slab pior over parameter of the network. The paper proposes Beta hyperpriors over the network, motivated by the Indian Buffet Process, and propose a method for input-conditional priors.

The paper is well-written and the material is communicated clearly. The topic is also of interest to the community and might have important implications down the road.

The authors, however, failed to convince the reviewers that the paper is ready for publication at ICLR. The proposed method is very similar to earlier work. The reviewers think that the paper is not ready for publication.